# Biallelic *PMS2* Mutations in a Family with Uncommon Clinical and Molecular Features

**DOI:** 10.3390/genes13111953

**Published:** 2022-10-26

**Authors:** Monica Pedroni, Maurizio Ponz de Leon, Luca Reggiani Bonetti, Giuseppina Rossi, Alessandra Viel, Emanuele Damiano Luca Urso, Luca Roncucci

**Affiliations:** 1Department of Medical and Surgical Sciences, University of Modena and Reggio Emilia, Via Del Pozzo 71, 41125 Modena, Italy; 2Division of Functional Onco-Genomics and Genetics, Centro di Riferimento Oncologico di Aviano, IRCCS, Via Gallini 2, 33081 Aviano, Italy; 3Surgery 3, Department of Surgical, Oncological and Gastroenterological Sciences, University of Padua, Via Giustiniani 1, 25128 Padua, Italy

**Keywords:** colon, cancer, *PMS2* gene, VUS, CMMR-D

## Abstract

We describe a patient with constitutional mismatch repair-deficiency (CMMR-D) in whom the syndrome started at age 10 with the development of multiple adenomas in the large bowel. In the successive 25 years, four malignancies developed in different organs (rectum, ileum, duodenum, and lymphoid tissue). The patient had biallelic constitutional pathogenic variants in the *PMS2* gene. We speculate that besides the *PMS2* genotype, alterations of other genes might have contributed to the development of the complex phenotype. In the nuclear family, both parents carried different *PMS2* germline mutations. They appeared in good clinical condition and did not develop polyps or cancer. The index case had a brother who died at age three of lymphoblastic leukemia, and a sister who was affected by sarcoidosis. Tumor tissue showed diffuse DNA microsatellite instability. A complete absence of immunoreactivity was observed for the PMS2 protein both in the tumors and normal tissues. Next-generation sequencing and multiple ligation-dependent probe amplification analyses revealed biallelic *PMS2* germline pathogenic variants in the proband (genotype c.[137G>T];[(2174+1_2175-1)_(*160_?)del]), and one of the two variants was present in both parents—c.137G>T in the father and c.(2174+1-2175-1)_(*160_?)del in the mother—as well as c.137G>T in the sister. Moreover, Class 3 variants of *MSH2* (c.1787A>G), *APC* (c.1589T>C), and *CHEK2* (c.331G>T) genes were also detected in the proband. In conclusion, the recognition of CMMR-D may sometimes be difficult; however, the possible role of constitutional alterations of other genes in the development of the full-blown phenotype should be investigated in more detail.

## 1. Introduction

Over the last 30 years, Lynch syndrome has been established as the most frequent form of inherited colorectal cancer, associated with 1–5% of all malignancies of the large bowel [1,2]. Undergoing surgery is the usual treatment for these patients, though evidence suggests that polyps and cancer might be prevented with long-term administration of anti-inflammatory drugs [3]. The syndrome is due to constitutional mutations in a class of genes whose function is to maintain genomic integrity (DNA mismatch repair genes, MMR) [4]. Some 70–90% of all cases are due to mutations in *MSH2* or *MLH1* genes. In the remaining cases, other less frequently mutated genes are involved, such as *MSH6*, *EPCAM,* or *PMS2* [5,6]. Studies of the *PMS2* gene’s pathogenic variants have been hindered by several technical difficulties, which have been overcome through the improvement of analytical procedures [7] and with the advent of next-generation sequencing (NGS) [8]. More recently, patients have shown biallelic (homozygous or compound heterozygous) alterations in one of the genes associated with Lynch syndrome [9]. The affected proband inherits one pathogenic variant from the paternal lineage and a second variant from the maternal lineage. The syndrome, known as “biallelic (or constitutional) mismatch repair-deficiency” (BMMR-D or CMMR-D), shows a particularly severe phenotype featuring early-onset malignancies of the alimentary tract, multiple adenomas, tumors in the nervous tissue, leukemia, lymphoma, and other neoplasms of the Lynch spectrum [10]. Profound impairment of the mismatch repair system leads to a lack of immunohistochemical expression of the protein corresponding to the mutated gene, both in tumors and normal tissues.

We describe a patient with constitutional biallelic pathogenic variants in the *PMS2* gene. The disease manifested early (age 10) with features of adenomatous polyposis (FAP/AFAP), and was complicated by the development of four different malignancies in the subsequent 20 years. It is possible that additional variants (Class 3) of the *APC, MSH2* and *CHEK2* genes contributed to the complex and severe phenotype. The four surviving members of the nuclear family were all carriers of *PMS2* pathogenic variants (biallelic in the proband, monoallelic in his parents and sister). The fifth family member (his brother) died at age 3 (1989) of lymphoblastic leukemia, and sarcoidosis was diagnosed in the proband’s sister.

## 2. Family Report

### 2.1. The Proband

The proband is a 40-year-old man of Italian ancestry, born in 1982 (174 cm, 70 kg) and evaluated for his clinical features in several institutions in Northern Italy. At age 10, rectal bleeding was noted. A first colonoscopy (1992) identified the presence of eight polyps (adenomatous at histology) between 5 and 10 mm in diameter, which were subsequently removed. The physical examination was normal; however, the patient had multiple freckles and several hyperpigmented spots on the thorax. Endoscopies were repeated in the following years and 2–10 adenomatous polyps with mild-to-moderate dysplasia were continually observed and removed. On one occasion, the pathologist described features of hamartoma in one of the lesions. Over time, the polyps observed using endoscopy tended to become larger (between 10 and 40 mm), especially in the sigmoid and descending colon. In 2007 (age 25), a colonoscopy showed more than 20 lesions (not otherwise specified) scattered from the rectum to the transverse colon, measuring between a few millimeters and 15–20 mm. In one of these polyps (in the sigmoid), features of adenocarcinoma infiltrating the muscularis mucosae were observed. The first gastroduodenoscopy (2007) was normal.

With a clinical suspicion of familial adenomatous polyposis or attenuated polyposis (FAP or AFAP), the patient underwent a total colectomy with ileo-anal anastomosis (2007). In the resected specimen, a T2N0Mx (Dukes’ A) carcinoma of the sigmoid was diagnosed together with an unspecified number of adenomatous lesions of various sizes (presumably less than 20). A molecular analysis of the *APC* gene showed an absence of truncating mutations; however, the variant c.1589T>C was detected, which caused a substitution of valine with alanine at position 530 of the encoded protein. This alteration should be considered as a variant of unknown significance since it has not been clearly associated with the FAP/AFAP phenotype. Analysis of the *MUTYH* gene showed a normal sequence. The patient was, therefore, considered to be a case of FAP/AFAP not associated with constitutional mutations of the major genes implicated in the syndrome (around 10% of all cases). However, further analysis of the resected carcinoma showed an elevated microsatellite instability with all markers, together with altered expression of the MLH1 protein in immunohistochemistry (MSH2 and MSH6 normally expressed, PMS2 not tested). Sanger sequencing and multiple ligation-dependent probe amplification (MLPA) analysis of the *MLH1* gene were normal. Moreover, the vague suspicion of type I neurofibromatosis led to a FISH analysis of the *NF1* gene, which did not identify any deletion. Due to the hamartomatous histological type of one of the removed colorectal polyps and the suspicion of Cowden disease [11], direct sequencing of the *PTEN* gene was also carried out without detecting any relevant variants. The patient was treated with anti-inflammatory compounds for nearly 2 years (sulindac, aspirin).

In 2009 (age 27), because of persisting dyspeptic symptoms, the patient underwent various endoscopic examinations, including enteroscopy. This revealed the presence of a substenosing lesion approximately 150 cm from the Treitz, which required jejunal resection of 8 cm. A histological examination identified a tubular adenoma with low-grade dysplasia. The patient was kept under close endoscopic control, which revealed frequent lesions in the stomach (of glandular histological type), duodenum and pouch (adenomas). In 2012 (age 30), a videocapsule investigation revealed the presence of a 2 cm lesion in the ileum. Due to difficulties in removing the mass through endoscopy, the patient was operated on again for an ileal resection. During an anatomical examination, an infiltrating adenocarcinoma was detected with a transmural extension through the ileal wall (T4N0Mx). The patient received fluorouracil + oxalyplatin chemotherapy for 6 months.

The proband maintained a relatively good condition for nearly 3 years. In 2016 (age 34), he complained about vague abdominal discomfort, weight loss and frequent vomiting. During a gastroscopy, a 2.5 cm polypoid lesion was noted in the second portion of the duodenum (1 year since the previous endoscopic control). Again, after attempts to remove the lesion endoscopically, the patient was operated on of duodenocefalopancreasectomy for a well-differentiated duodenal adenocarcinoma (T3N0Mx).

With the advent of NGS, genomic DNA of the patient was analyzed with a panel of 27 genes, many of which related to colorectal malignancies (432 target regions; 104.776bp: *APC*, *ATM*, *BARD1*, *BRCA1*, *BRCA2*, *BRIP1*, *CDH1*, *CHEK2*, *EPCAM*, *FAM175A*, *MLH1*, *MRE11A*, *MSH2*, *MSH6*, *MUTYH*, *NBL*, *PALB2*, *PIK3CA*, *PMS2*, *PMS2CL*, *PTEN*, *RAD50*, *RAD51C*, *RAD51D*, *STK11*, *TP53*, *XRCC2*). The results of molecular investigations are shown in Table 1A.

The missense variant c.137G>T of the *PMS2* gene, exon 2 (Figure 1), causes the substitution of the amino acid serine with isoleucine at position 46 of the encoded protein, (p.Ser46Ile). According to the present knowledge, this is considered a Class 4 mutation [12]; it has been associated with Lynch syndrome and reported in databases (ClinVar and LOVD).

The confirmed missense variant in *APC* and the newly detected *CHEK2* and *MSH2* variants were all interpreted as variants of uncertain significance (VUS), not definitively associated with the Lynch or FAP phenotype. The missense variants of *APC* and *MSH2* were already reported in the databases. Only the *CHEK2* variant was unreported. The MLPA analysis revealed the presence of a second mutation in the *PMS2* gene, an ample genomic deletion removing exons 13 to 15 of the gene encompassing nucleotides 2175 to 2589 (Figure 2). This variant is a never-before-described Class 5 mutation associated with Lynch syndrome in heterozygosis. The two variants of *PMS2* identified in the proband as compound heterozygosis are responsible for CMMR-D. The immunohistochemistry showed a lack of PMS2 expression both in the tumor masses previously resected, and the surrounding normal ileal and jejunal mucosa (Table 1B).

In December 2017 (age 35), in absence of relevant symptoms, a routine chest X-ray showed an enlarged mediastinum. Further examinations (Computed Tomography [CT] scan, Positron Emission Tomography [PET] scan, and biopsies) led to the diagnosis of a mediastinal tumor that had presumably originated in the thymus. The patient was operated on and a 3.5 cm × 2.0 cm × 1.8 cm mass was removed. During histology, a proliferation of thymic elements together with T lymphocytes was observed. Reaching a proper diagnosis was difficult and required consultations with various experts. After the resection and examination of an enlarged sopraclavear lymph node and further characterization of the cellular elements, an agreement was reached on the definitive diagnosis of lymphoblastic lymphoma. In February 2018, the patient started chemotherapy (eight cycles of various combinations of vincristine, asparaginase, cyclophosphamide and methotrexate, followed by radiotherapy of the mediastinum), which was well-tolerated. During his last visit (June 2021), the patient appeared to be in a good clinical condition. On that occasion, the MLPA analysis revealed the presence of a second mutation in the *PMS2* gene, described above.

### 2.2. The Family

The family of the proband (Figure 3) shows some features of interest. Three subjects from the first generation (grandfathers and grandmothers of the proband) are still alive and have not reported relevant diseases. The fourth died at age 80 from uncertain causes. The proband’s parents, born in 1955 and 1953, have not reported significant pathologies; both underwent colonoscopies around age 50, which were unremarkable. The proband’s brother, born in 1986, died at age three (1989) of lymphoblastic leukemia. His clinical charts and further details were not available.

In 2018, the proband’s sister (III-3, age 28) underwent an abdominal ultrasound owing to vague abdominal discomfort. This revealed several 1–2 cm nodules in the spleen. At age 16 and 20, she had two normal colonoscopies, and a CT scan and a Nuclear Magnetic Resonance (NMR) of the abdomen and chest confirmed multiple nodules (0.5–1.5 cm) of the spleen, which was slightly enlarged. Small nodules were seen in the liver, and the mediastinum appeared moderately enlarged because of several slightly prominent lymph nodes. A PET scan also revealed increased captation of the lesions. A fine-needle biopsy of the spleen was executed and a granulomatous pattern was observed at histology, with non-necrotizing sarcoidosis-like lesions, giant cells with multiple nuclei, eosinophilic granulocytes, and low lymphoid infiltrate. A search for bacteria and protozoa (including Mycobacterium tuberculosis) was negative. A diagnosis of sarcoidosis was proposed, and blood tests were all in the normal range. In 2021, the parents and sister agreed to be tested for a targeted variant analysis of *PMS2* and other relevant genes (Table 1A). It should be noted that other genetic variants could have been present in family members, especially those involved in the onset of sarcoidosis, but they were not searched for.

## 3. Molecular Investigations

Following the first surgical operation (2008), the proband was actively investigated for confirming or excluding the diagnosis of Lynch syndrome or FAP/AFAP. In the tumor tissue, microsatellite instability (MSI) with various markers and immunohistochemical expression of mismatch repair genes was investigated with standard procedures [13,14,15]. After the extraction of genomic DNA from peripheral white blood cells, direct sequencing of the *APC*, *MUTYH* and *MLH1* genes was carried out with the Sanger technique as already reported [16,17]. Additionally, MLPA for the same genes was used to search for possible large deletions/duplications [17]. The Sanger sequencing of exons 1–9 and flanking regions of the *PTEN* gene was also carried out and a FISH analysis was executed to detect possible deletions of the *NF1* gene [18].

In 2017, with the advent of NGS, the genomic DNA of the patient was evaluated with the Hereditary Cancer Solution (HCS, SOPHIA genetics, Saint Sulpice, Switzerland), which included 27 genes on a target region of 105 Kb. The HCS panel covered the coding sequences and splicing junctions of most clinically relevant genes associated with cancer risk, some of which were related to colorectal tumors [8]. The library was prepared using capture-based target enrichment sequenced on a Miseq Platform (Illumina, Eindhoven, The Netherlands) and analyzed on the SOPHIA DDM^®^ platform according to the manufacturer’s protocols. The confirmation of the sequence variants with unknown or possible pathogenic significance was obtained through a single PCR and direct Sanger sequencing. Finally, a large deletion/duplication analysis of the *PMS2* gene was integrated using an updated version of the MLPA assay. All detected DNA variants were interpreted according to the InSiGHT classification criteria [19] and the American College of Medical Genetics and Genomics (ACMG) guidelines [20].

## 4. Discussion

We described a patient with CMMR-D caused by biallelic *PMS2* mutations in a nuclear family with rather peculiar clinical features. *PMS2* germline alterations were found in all four living family members. Among his siblings, the proband (and presumably the brother, deceased in 1989) was the only living subject who inherited the two deleterious alterations from his parents and developed the full-blown phenotype. The patient—a 40-year-old man at present in good clinical condition—developed four independent malignancies (adenocarcinomas in the sigmoid colon, jejunum, duodenum, and a lymphoblastic lymphoma) within 10 years, together with features of attenuated polyposis of the large bowel that led to proctocolectomy and ileoanal anastomosis at age 25. In addition to the *PMS2* gene, molecular investigations led to the detection of VUS in the *APC, MSH2,* and *CHEK2* genes, which might also be involved in polyps and tumor development.

One area of interest with this patient (and his family) was the difficulty in reaching a proper diagnosis. From the beginning of his clinical history (1992, age 10) to proctocolectomy (2007, age 25), a diagnosis of FAP/AFAP seemed the most likely, though genetic testing—limited to *APC* and *MUTYH* genes—was negative. The continuous reoccurrence of newly developed polyps, especially in the left colon (up to “more than 20”), led to an attempt at medical treatment using sulindac, but with limited results. Subsequently, the finding of hamartomatous lesions opened up new scenarios, such as possible Cowden disease or type 1 neurofibromatosis; however, those hypotheses were not supported by the molecular findings. When Lynch syndrome was considered, the *MLH1* gene was suspected because of a lack of expression in the encoded protein, presumably due to an artefact. However, gene sequencing did not reveal any abnormality in *MLH1*. PMS2 immunohistochemistry was not initially executed, which was another reason for diagnostic delay, although for *PMS2,* this is considered to be less informative than for other mismatch repair genes [21,22].

The frequency of *PMS2* mutations among patients with Lynch syndrome is around 5%, and studies on these have been hampered by technical problems, such as the presence of pseudogenes localized in the same chromosome [23]. In *PMS2* families, the penetrance of mutations is lower than major Lynch syndrome-associated genes (*MSH2* and *MLH1*), with a 0-70 year-cumulative incidence of 15–20% for cancer of the large bowel and 15% for endometrial tumors [24]. The clinical spectrum of neoplasms does not seem different from the classical Lynch syndrome, with colorectal and endometrial cancer accounting for about 60% of the total tumors together with rather frequent small bowel tumors [25]. Considering the low frequency of *PMS2* mutations, reduced penetrance, and a limited value for immunohistochemistry, it is not surprising that the correct diagnosis of *PMS2*-associated Lynch syndrome is, in many cases, more difficult than for the two major genes.

Different from patients carrying heterozygous germline alterations, patients with biallelic *PMS2* mutations (as well as a biallelic state of other genes involved in Lynch syndrome) have a constitutional deficiency in the mismatch repair system, leading to a generalized defect of microsatellite replication in all tissues. This condition translates to an early onset of frequently aggressive neoplasms that are not limited to the gastrointestinal tract but involve the nervous tissue, hematologic malignancies, and tumors of other organs [26,27,28]. In these families, history of cancer may be lacking, owing to a low penetrance of *PMS2* heterozygous mutations and frequent early death of homozygotes.

Our proband meets the clinical criteria for CMMR-D syndrome entirely, with the development of several malignancies located in various tracts of the intestine starting at age 25 and a lymphoblastic lymphoma at age 35. Moreover, multiple adenomatous polyps were found when the patient was young (age 10), and the clinical situation prompted the execution of a colectomy with ileoanal anastomosis. The family is also a typical case of CMMR-D, without a vertical transmission of cancer, despite both parents being carriers of deleterious (Class 4–5) mutations in the *PMS2* gene. The proband’s brother died at age three of lymphoblastic leukemia and he presumably carried the same compound heterozygosis of the index case, though his early death prevented subsequent development of the full-blown phenotype.

It cannot be excluded “a priori” that constitutional alterations of other genes might have contributed to the development of the proband’s phenotype. In addition to his biallelic *PMS2* mutations, which were already described in the main databases (LOVD, ClinVar, UMD), germline missense variants in three other genes were detected. The *APC* variant c.1589T>C was classified as Class 3 (VUS) [19]. Although not reported in the main databases as clearly associated with the FAP/AFAP phenotype, it could be speculated that its presence in the proband and a possible, though unproven, interaction with one or more of the other genetic abnormalities could be an additional factor that was responsible for the numerous adenomatous polyps observed in the patient.

The *MSH2* alteration c.1787A>G not associated with a lack of expression of the corresponding protein was similarly classified as Class 3 and never referred to the Lynch phenotype. Its effect might have been to reinforce the oncogenic potential of the primary *PMS2* biallelic mutation. Finally, the *CHEK2* variant c.331G>T was again a Class 3 alteration. *CHEK2* was reported to be a moderate-penetrance, multi-organ cancer susceptibility gene, mutations of which increase the risk of various tumors, such as neoplasms of the breast, large bowel, and prostate [29]. Moreover, it was reported that variants of *CHEK2* could modify the risk and clinical course of non-Hodgkin lymphoma [30]. The digenic inheritance hypothesis was tested by running DIVAs (enGenome srl, Pavia, Italy), which is a machine-learning-based method for identifying and classifying digenic variants. Briefly, this method, starting from the patient’s phenotypes (expressed as HPO terms) and variants, classifies each digenic combination as pathogenic or benign. A score is assigned to each combination, with values between zero and one. The higher the score, the more evidence of causation exists for the combination. All possible digenic combinations, including the *PMS2* c.137G>T point mutation, were classified as pathogenic by DIVAs, reinforcing our supposition with scores ranging from 0.99 to 1.00. Unfortunately, the *PMS2* large deletion could not be tested by this method, preventing us from drawing a complete picture of the family genotypes.

While we have already discussed the proband’s brother, the parents are presently in their sixties and in good health; their endoscopic controls were negative despite the fact that they are both carriers of *PMS2* mutations. This, however, is not surprising, given the low penetrance of the gene in heterozygosity [31]. The proband’s sister (III-3, Figure 3), a *PMS2* mutation carrier, had histologically documented sarcoidosis. To our knowledge, the disease does not bear any relationship with Lynch syndrome, mismatch-repair deficiency, or alterations in the investigated genes.

## 5. Conclusions

In conclusion, we believe that the index case and the family described in the report allow us to contribute some points of value. First, the diagnosis of CMMR-D can be difficult in certain circumstances, especially when the patient is evaluated in different institutions. Second, it is possible, though unproven, that alterations of several cancer-related genes might contribute to the full clinical expression of the disease. Third, the advent of new technologies, such as NGS, should facilitate the diagnosis and might reveal possible interactions among genes.

## Figures and Tables

**Figure 1 genes-13-01953-f001:**
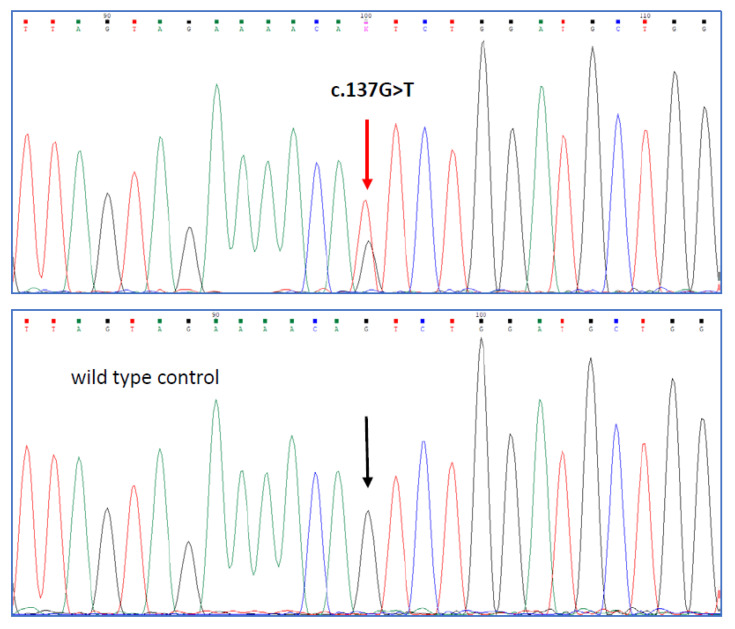
Sanger sequencing. Electropherograms of proband (**top**) and control (**bottom**) showing the c.137G>T substitution in exon 2 of the *PMS2* gene.

**Figure 2 genes-13-01953-f002:**
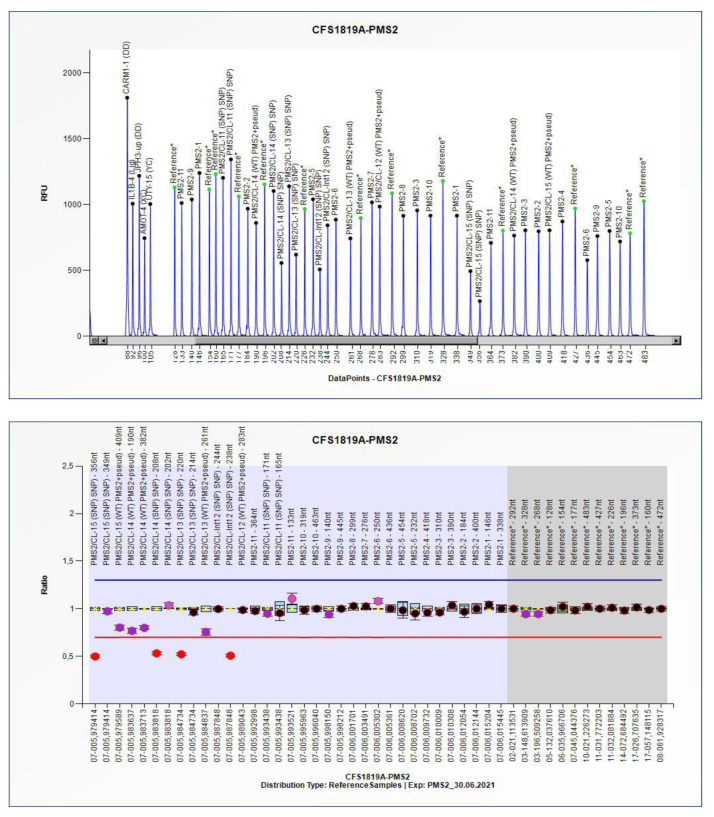
Multiplex Ligation-dependent Probe Amplification (MLPA) analysis. Electropherogram ratio chart showing the copy number changes in the *PMS2/PMS2CL* genes. This sample report, generated by the software Coffalyzer.net (MRC Holland), shows the heterozygous deletion c.(2174+1_2175-1)_(*160_?)del spanning from intron 12 to exon 15 of the *PMS2* gene.

**Figure 3 genes-13-01953-f003:**
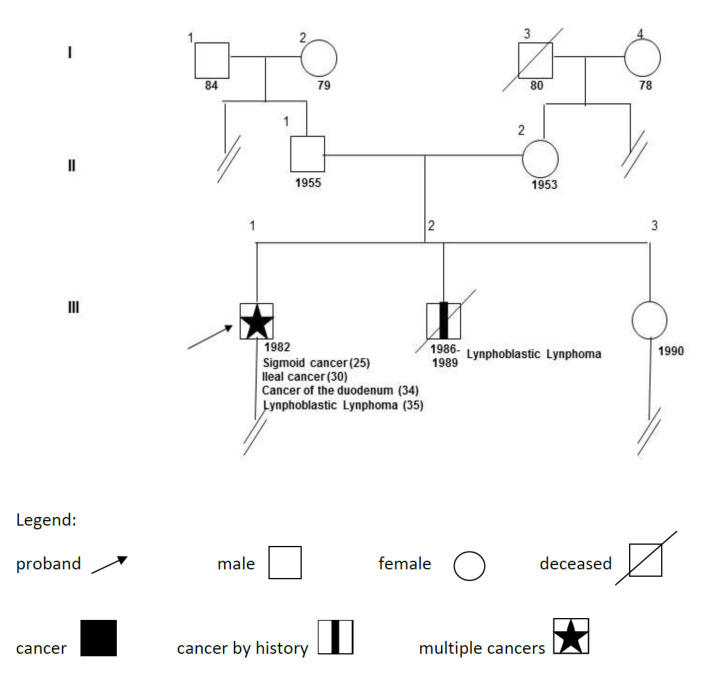
Genealogical tree of the family. The proband (III-1, arrow), born in 1982, had 4 cancers whose ages at diagnosis are reported in brackets.

**Table 1 genes-13-01953-t001:** Genetic alterations detected in the family by DNA analyses (**A**) and immunohistochemistry (**B**).

(A)					
**Constitutional DNA**	** *PMS2* **		** *APC* **	** *MSH2* **	** *CHEK2* **
Proband (2008, 2018, 2021)	c.137G>T p.(Ser46Ile)		c.1589T>C p.(Val530Ala)	c.1787A>G p.(Asn596Ser)	c.331G>T p.(Asp111Tyr)
	c.(2174+1_2175−1)_(*160_?)del p.?				
Father (2021)	c.137G>T			c.1787A>G	c.331G>T
Mother (2021)	c.(2174+1_2175−1)_(*160_?)del p.?		c.1589T>C		
Sister (2021)	c.137G>T			c.1787A>G	
**(B)**					
**Tumor Tissue**					
**Jejunum (2012)**		PMS2 *		not expressed	
		MLH1		not expressed	
		MSH2, MSH6		Expressed	
**Duodenum (2016)**		PMS2 *		not expressed	
		MLH1, MSH2, MSH6		Expressed	

* Lack of PMS2 expression both in the tumor mass and in the normal surrounding mucosa.

## Data Availability

Not applicable.

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
