# Peer review of "Biallelic PMS2 Mutations in a Family with Uncommon Clinical and Molecular Features"

_genes, 2022, doi:10.3390/genes13111953_

Round 1

Reviewer 1 Report

To the authors:

The report describes a clinical case of Constitutional Mismatch-Repair Deficiency which is very interesting due to the severe phenotype of the subject. This is a welcome contribution to the scientific literature as it is a case with a very long follow-up and with both a therapeutic and diagnostic approach described in detail over the years.  I think it is a very good case for publication, with an unusual genotype that invites a broader diagnostic approach when typical mismatch repair markers are not altered but the clinical signs support a diagnosis of Mismatch-Repair Deficiency. However, several points could be improved or completed.

1   1. Although the progression of the case is generally well understood, it would help reading comprehension to include the date on which the mutations were determined in the different family members in table 1.A, as is done for the immunohistochemical markers in table 1.

2      2. I recommend checking the spelling as there are a few errors. Among them, in lines 150-155 the text of lines 132-137 is repeated.

3      3. I think it is interesting to mention that both the mutations found and the PMS2 deletion have been described before (check ClinVar databases, LOVD...), and in the case of APC and MSH2 alterations in several individuals with cancer predisposition syndromes. Since it gives robustness to the hypothesis that these other alterations could be the cause of the severe phenotype associated with the PMS2 biallelic alteration.

4       4. I miss illustrative images of the genetic findings such as the electropherogram of the alterations in the proband, as well as the image of the PMS2 MLPA deletion.

5      5. Regarding the NGS panel with which the alterations in the other affected genes were discovered, I recommend adding the list of the 27 genes included. In this way, is easier to understand the candidate genes tested by reading the article directly. Thus, the reader can see that the genes involved in sarcoidosis, the condition affecting the proband's sister, would not be included (CARD15, NOD2...). I also find it interesting to mention that there may be more genetic alterations involved in the family context that have been inherited by the sister and have not been tested for.

Reviewer 2 Report

This is an interesting cancer genetics case study, which is very well presented. I don’t have any major issues with this paper.

My minor comments/questions is about PMS2 IHC in the proband and underlying reasons.

1.        It would be nice (not imperative) to present IHC data for PMS2 protein from the Mother (with the deleterious mutation) and Father (with the VUS), who had colonoscopy (Biopsy may be taken during their colonoscopies) for comparison purposes. Considering, normal tissue does not express PMS2 in the proband, and (assuming IHC done properly with external controls), total loss of expression of PMS2 protein must be due to PMS2constitutional mutation(s). To what extent PMS2 expression is retained in the Mother and Father can be help elucidating pathophysiology

2.       Presentation of somatic NGS studies (or at least PMS2) gene studies in the proband, would be also understand the ongoing pathophysiology (though I understand it is not main focus of the paper)
